# Outcomes and Predictors of In-Hospital Mortality among Older Patients with Dementia

**DOI:** 10.3390/jcm12010059

**Published:** 2022-12-21

**Authors:** Giuseppe De Matteis, Maria Livia Burzo, Davide Antonio Della Polla, Amato Serra, Andrea Russo, Francesco Landi, Antonio Gasbarrini, Giovanni Gambassi, Francesco Franceschi, Marcello Covino

**Affiliations:** 1Department of Internal Medicine, IRCCS Fondazione Policlinico Universitario Agostino Gemelli, 00168 Rome, Italy; 2Department of Internal Medicine, Ospedale Santo Spirito in Sassia, 00193 Rome, Italy; 3Emergency Department, IRCCS Fondazione Policlinico Universitario Agostino Gemelli, 00168 Rome, Italy; 4School of Medicine, Università Cattolica del Sacro Cuore, Largo Francesco Vito 1, 00168 Rome, Italy; 5Department of Aging, Neurological, Orthopaedic and Head-Neck Sciences, IRCCS Fondazione Policlinico Universitario Agostino Gemelli, 00168 Rome, Italy; 6Dipartimento di Medicina e Chirurgia Traslazionale, Università Cattolica del Sacro Cuore, 00168 Rome, Italy

**Keywords:** dementia, older patients, respiratory failure, sepsis, mortality

## Abstract

Dementia is associated with high rates of admission to hospital, due to acute illness, and in-hospital mortality. The study aimed to investigate the impact of dementia on in-hospital mortality and identify the predictors of in-hospital mortality in these patients. This was a retrospective study evaluating all the patients ≥65 years consecutively admitted to our Emergency Department (ED). We compared the clinical outcomes of the patients with dementia at ED admission with those who did not have dementia, using a propensity score-matched (PSM) paired cohort of controls. The patients were matched for age, sex, Charlson Comorbidity Index value, and clinical severity at presentation (based on NEWS ≥ 5). The primary study endpoint was all-cause in-hospital death. After the PSM, a total of 7118 patients, 3559 with dementia and 3559 in the control group, were included in the study cohort. The mean age was 84 years, and 59.8% were females. The overall mortality rate was higher for the demented patients compared with the controls (18.7% vs. 16.0%, *p* = 0.002). The multivariate-adjusted hazard ratio (HR) showed that dementia was an independent risk factor for death (HR 1.13 [1.01–1.27]; *p* = 0.033). In the patients with dementia, respiratory failure (HR 3.08 [2.6–3.65]), acute renal failure (HR 1.64 [1.33–2.02]; *p* < 0.001), hemorrhagic stroke (HR 1.84 [1.38–2.44]; *p* < 0.001), and bloodstream infection (HR 1.41 [1.17–1.71]; *p* = 0.001) were significant predictors of worse outcomes. Finally, the comorbidities and severity of illness at ED admission negatively influenced survival among the patients with dementia (CCI HR 1.05 [1.01–1.1] *p* = 0.005; NEWS ≥ 5 HR 2.45 [1.88–3.2] *p* < 0.001). In conclusion, among the hospitalized older patients, dementia was associated with a higher risk of mortality. Furthermore, among the older patients with dementia, respiratory failure and bloodstream infections were independently associated with an increased risk of in-hospital mortality.

## 1. Introduction

Dementia is a complex clinical syndrome characterized by progressive cognitive decline and associated with a deterioration in behavior and reduction of the capacity for independent living [1]. It includes a group of related neurodegenerative pathologies such as Alzheimer’s disease (AD), vascular dementia, Lewy body dementia, frontotemporal dementia, and Parkinson’s disease dementia.

Currently, the prevalence of dementia is estimated to be around 5% of the population [2] and 51.8% of nursing home residents [3]. However, dementia typically affects older patients and, largely due to the increase in population growth and the progressive population aging, a significant increase in the number of individuals affected by dementia can be expected [4].

In patients with dementia, high mortality rates are reported [5]. In particular, older-onset (>65 years of age), male sex, the neurodegenerative pathology involved and its severity, impact on autonomy in daily life, and the consequent reduction in cognitive performance have been associated with this increase in mortality [6,7,8,9,10,11].

Admission to the hospital due to acute illness, including Emergency Department (ED) admission, is common for people affected by dementia, especially in their last years of life, representing a stressful event for older patients and their caregivers [12,13]. The most frequent causes of acute hospitalization are infectious diseases [14,15], and the presence of dementia has been associated with an increased rate of hospitalization-related infections and severe sepsis, with dementia itself being the cause of admission in a minority of cases [16,17].

In patients with dementia, the rate of comorbidities also increases with age [5,18,19], and the association between dementia and comorbidities, mainly cerebrovascular disease, diabetes, and hypertension, increases the risk of adverse outcomes [8,20]. In addition, dementia increases the hospital length of stay (LOS) [21,22], with both cognitive and physical impairment and with a significant impact on mortality [12,14].

To date, despite the high prevalence of older patients hospitalized who are affected by dementia, the effect of dementia on mortality is often underestimated, and studies investigating the impact of dementia on prognoses are still lacking [8,23,24,25].

Thus, our study aimed to describe the clinical characteristics of older patients affected by dementia and hospitalized, investigate the impact of dementia on in-hospital mortality, and identify the predictors of in-hospital mortality in these patients.

## 2. Materials and Methods

### 2.1. Study Population

This is a single-center, retrospective study conducted in a large university hospital [Fondazione Policlinico Universitario A. Gemelli IRCCS, Rome]. We enrolled all the patients over the age of 65 who had been diagnosed with dementia, admitted to ED, and then hospitalized, over a six-year period from 1 January 2014 to 31 December 2019.

We included in the study sample patients who had received a definitive diagnosis of dementia. The judgment criteria included both an admission diagnosis of dementia and the presence of dementia in the hospital discharge record. The diagnosis at hospital discharge was based on the codes defined by the “International Classification of Disease, tenth revision” (ICD-10 CM). However, as is more extensively described in the limitations section of the study, given the retrospective nature of our analysis, the data about the subtype of dementia and its severity were not available.

The patients presenting to the ED and admitted to an intensive care unit (ICU) were excluded from the study. For the patients > 65 years with multiple, recurrent ED admissions, only the data from the first hospitalization were considered.

We compared the clinical outcome of the patients with dementia at ED admission with those who did not have dementia, using a propensity score-matched (PSM) paired cohort of controls. The patients were matched for age, sex, Charlson Comorbidity Index value, and clinical severity at presentation (based on NEWS ≥ 5).

### 2.2. Study Variables

The data were obtained through consultation of electronic medical records, and each patient’s health record was used to collect demographic and clinical characteristics, the data regarding ED presentation, and events occurring during the hospital stay, including the outcome at discharge. The clinical records were also reviewed to assess comorbidities based on the patient’s history and hospital discharge diagnosis.

We evaluated the following information:Demographic data: age and sex.Clinical presentation at the admission to ED: dyspnea, fever, chest pain, syncope, abdominal pain, diarrhea, vomiting, malaise/fatigue.Physiological parameters at ED admission: body temperature, heart rate, respiratory rate, systolic blood pressure, and level of consciousness assessed by the response on the AVPU (Alert, Voice, Pain, Unresponsive) scale, peripheral oxygen saturation (SpO_2_) in ambient air. Based on these parameters, the national early warning score (NEWS) was calculated, and we categorized patients’ illness severity at admission as NEWS < 5 or NEWS ≥ 5 [26].Clinical history and comorbidities: ischemic heart disease (IHD), congestive heart failure (CHF), peripheral artery disease (PAD), cerebral vascular disease (previous stroke), diabetes, chronic liver disease, chronic obstructive pulmonary disease (COPD), chronic kidney disease (CKD). Overall comorbidity presence and severity were assessed by the Charlson Comorbidity Index (CCI). The CCI is a validated score that takes into account the number and the seriousness of the comorbid disease and assigns weights, from 1 to 6, to each comorbidity for every individual. A CCI score of zero represents no comorbidities, and a score from 1 to ≥6 represents a gradually higher load of comorbidities, with a corresponding increase in mortality [27].

Moreover, acute medical conditions, defined as a new onset or acute exacerbation of a preexisting clinical condition, were assessed: respiratory failure (here including any acute medical condition needing oxygen therapy), COPD exacerbation, acute heart failure, acute myocardial infarction, acute atrial fibrillation, pulmonary embolism, ischemic and hemorrhagic stroke, bloodstream infections, pneumonia, abdominal infection, urinary tract infection, and acute kidney injury (AKI).

### 2.3. Outcome Measures

The primary endpoint was all-cause in-hospital death.

Secondary endpoints were the identification of predictors of in-hospital mortality in patients affected by dementia and the assessment of the role of dementia on the average LOS.

### 2.4. Statistical Analysis

The categorical variables were presented as numbers and percentages. The continuous normally distributed variables were presented as the mean ± standard deviation; the non-normally distributed data were presented as the median [inter-quartile range], and the binary or ordinal variables were presented as absolute frequency (%).

The sample was divided into two groups, a study group of patients with a known diagnosis of dementia, and a control group obtained by matching a cohort of controls by 1:1 propensity score matching (PSM). The PSM was calculated by a logistic regression model using the nearest neighbor technique. The variables considered for the PSM were: age, gender, comorbidities, and NEWS score > 5 [25,26]. An optimal matching with a caliper size of 0.2 was used to avoid poor matches. A description of the PSM analysis and distribution before and after the match is provided in the Appendix A.

The LOS was calculated from the time of ED admission to the discharge or death. The survival analysis was performed according to the Kaplan–Meier methods, and the differences in survival were assessed using the log-rank test. The study variables were assessed for the association with all-cause in-hospital death by a univariate Cox regression analysis. The significant variables in the univariate analysis were entered into a multivariate Cox regression model to identify the independent risk factors for survival. A further analysis using the Cox model was performed by evaluating the clustered acute medical conditions. The association of factors with in-hospital death in the multivariate analysis is expressed as the hazard ratio (HR) [95% confidence interval].

A two-sided *p*-value of 0.05 or less was considered significant. All the data were analyzed by SPSS v26^®^ (IBM, Armonk, NY, USA).

## 3. Results

### 3.1. Study Cohort and Baseline Characteristics

Overall, 48,962 patients ≥ 65 years were evaluated at ED admission during the study period (Table 1). Among them, 3559 (7.3%) had a confirmed diagnosis of dementia. The people affected by dementia admitted to the ED were significantly older, with a median age of 85 years vs. 78 years compared to the entire population analyzed. The most common causes of ED access among all the patients were dyspnea (20.1%) and fever (19.0%), while syncope (7.7%) and diarrhea (3.4%) were less common. The most prevalent comorbidities were IHD (22.4%), PAD (22.0%), and diabetes (21.6%).

After the PSM, a total of 7118 patients, 3559 with dementia and 3559 in the control group, were included in the study cohort.

### 3.2. Comparison of the Matched Study Groups

The patients included in the study had a median age of 84 years and were predominantly women (59.8%). At admission, the patients had similar clinical presentations and severities, as defined by NEWS, in both groups (Table 2). Fever tended to be slightly more common in the demented patients but was not statistically significant (20.7% vs. 22.0%; *p* = 0.107). The time spent in the ED was less for the patients with dementia, with a median time of 21.3 h compared with 21.6 h in the controls.

Most of the enrolled patients had comorbidities, with a median CCI of 7. The patients with dementia had a more frequent history of stroke (23.4% vs. 17.2%), PAD (37.6% vs. 29.2%), and COPD (15.2% vs. 13.2%).

In the patients with dementia, IHD (18.1% vs. 24.9%), chronic liver disease (0.9% vs. 2.1%), and CKD (22.4% vs. 26.1%) were significantly rarer than in the non-demented. The comorbidities such as diabetes and CHF were balanced between the two groups.

The most common acute medical conditions affecting the patients in the study cohort were respiratory failure (19.3%), acute heart failure (17.8%), acute atrial fibrillation (17.5%), and pneumonia (17.1%).

The people affected by dementia were found to be more commonly hospitalized with infectious diseases, including mostly pneumonia (20.6%), urinary tract infections (13.4%), bloodstream infections (10.5%), and abdominal infections (2.9%). The dementia patients also showed a higher rate of hemorrhagic stroke, while no difference in the ischemic stroke rate was observed. Acute cardiovascular events were significantly more frequent in the control group, with a statistically significant difference in the case of acute myocardial infarction (3.7% vs. 2.3%; *p* = 0.001) and acute atrial fibrillation (19% vs. 16%; *p* < 0.001). No differences were found in terms of pulmonary embolism and AKI, while a slightly longer hospitalization related to respiratory failure (19.6% vs. 19.1%) and COPD exacerbation (5.0% vs. 4.1%) was observed in the control group.

By the end of the observational study period, 1237 patients (17.4%) had died, with a higher rate in the people affected by dementia (18.7% vs. 16.0%).

The mean LOS was 12.4 days, but the demented people had a slightly longer LOS compared with the controls (12.6 days vs. 12.2 days). In contrast, the rate of discharge home was significantly higher in the patients without dementia (69.2% vs. 60.8%).

### 3.3. Variables Associated with the Primary Endpoint (Survival Analysis in Patients with Dementia vs. Controls)

Overall, 1237 patients in the study died (Table 3). The deceased patients were more frequently women (54%), older (median age 81 years), and affected by dementia (53.8% vs. 49.2%; *p* < 0.001). A more severe clinical presentation and a higher deterioration of physiological parameters at ED admission, expressed by a NEWS ≥ 5, were associated with a significant increase in mortality (29.2% vs. 11.0%; *p* < 0.001).

The deceased patients were more frequently hospitalized for dyspnea (28.6% vs. 18.4%) and fever (23.4% vs. 20.9%), while admission for chest pain (4.8% vs. 2.1%) and syncope (11.1% vs. 5.1%) was the most common in the surviving group.

The presence of comorbidities also played a role in influencing mortality: the deceased patients showed a higher comorbidity rate (CCI = 6) compared with the surviving patients (CCI = 5). The deceased patients had a more frequent anamnesis of CHF (36% vs. 24.5%) and CKD (32.8% vs. 22.4%). No statistically significant differences were found in the rate of IHD, PAD, diabetes, and chronic liver disease among the deceased and surviving patients.

Furthermore, the deceased patients showed a higher frequency of respiratory failure (47.9% vs. 13.3%), acute heart failure (24.3% vs. 16.5%), sepsis (19.9% vs. 6.4%), pneumonia (28.1% vs. 14.7%), acute renal failure (18.1% vs. 8.6%), and hemorrhagic stroke (8.9% vs. 4.9%). COPD relapses (5% vs. 2.7%), as well as abdominal infections (2.8% vs. 1.1%) and urinary tract infections (11.3% vs. 6.2%), were more common in the surviving patients. The LOS was significantly shorter for the group of deceased patients (8.3 vs. 9.1).

### 3.4. Multivariate Analysis for In-Hospital Death

In the multivariate Cox regression analysis, several factors emerged as independent predictors of worse outcomes in our cohort (Table 4). Among these, dementia resulted in a significant independent risk factor for death (HR 1.13 [1.01–1.27]; *p* = 0.033). Predictably, a high rate of comorbidities and more severe clinical conditions on ED admission (NEWS ≥ 5) were associated with a worse prognosis in the whole sample.

Once adjusted for the baseline covariates (Table 5), the occurrence of respiratory failure was associated with an increased HR for death in the patients affected by dementia (HR 3.08 [2.6–3.65]).

Similarly, AKI (HR 1.64 [1.33–2.02]; *p* < 0.001), hemorrhagic stroke (HR 1.84 [1.38–2.44]; *p* < 0.001), and bloodstream infections (HR 1.41 [1.17–1.71]; *p* = 0.001) were other significant risk factors for mortality in the demented patients.

Finally, the comorbidities and severity of illness at ED admission negatively influenced survival among the patients with dementia (CCI HR 1.05 [1.01–1.1] *p* = 0.005; NEWS ≥ 5 HR 2.45 [1.88–3.2] *p* < 0.001).

## 4. Discussion

The main finding of the present study is that, among older patients admitted to the ED for any cause and then hospitalized, dementia is a risk factor for in-hospital mortality, regardless of the burden of other comorbidities and disease severity at hospital admission.

Dementia is a global health matter with a significant economic and social impact. Moreover, the aging of the worldwide population [28] is driving an unprecedented increase in the number of people with dementia. As a consequence, in recent decades, an emergent growth of acute hospitalizations in this population has been observed. Therefore, a better understanding of the factors affecting the clinical outcomes is crucial to improve the management of hospitalized patients with dementia.

In this study, we compared the clinical characteristics and the outcomes of a cohort of hospitalized older patients with and without dementia. Consistent with the previous reports, as well as the future trends depicted by the latest analysis, the demented patients in our cohort were older and more commonly females [3]. Moreover, they had a great burden of comorbidity, particularly heart failure, CKD, PAD, and a history of stroke, compared with the non-demented patients [29]. By contrast, as previously reported [30], chronic ischemic heart disease showed a lower prevalence, due to the increased lethality of this condition.

Furthermore, the present analysis showed that, among the demented patients, a high comorbidity burden and a more severe clinical presentation and ED admission were associated with a worse clinical outcome. To date, increasing age and dementia, as well as the number of comorbidities and worse clinical conditions at the time of ED admission, have already been recognized as risk factors for in-hospital mortality [31,32]. A large, registry-based cohort study conducted in a Danish population by Taudorf et al., investigating the impact of somatic and psychiatric comorbidities on mortality in patients affected by dementia aged ≥ 65 years, showed that mortality in the demented patients was more than double compared with the general population, even after adjusting for comorbidities. It suggested that dementia, per se, contributes to increased mortality, which is further increased by the comorbidities burden [20]. Similarly, our results were in line with those reported by Tonelli et al. [8], who showed that dementia and multimorbidity were strongly correlated with all-cause hospital admissions and increased in-hospital mortality. Furthermore, in one of the most comprehensive studies examining the trends in Alzheimer’s disease (AD) patient admissions, conducted by Beydoun et al. [29], having more than five comorbidities was associated with a more-than-doubled risk of mortality. Interestingly, among the comorbid conditions included in the Beydoun cohort, those which predicted mortality with the highest odds were CHF, renal failure, and peripheral vascular disorders.

Not surprisingly, given the intrinsic frailty of this patient population, adverse clinical events occurring during the hospital stay were correlated with a high risk of mortality [33]. As a consequence, the timely diagnosis and treatment of acute illness should represent a target for mortality prevention in these patients.

In the present study, we found that the demented patients showed a higher prevalence of infections, mainly pneumonia, followed by bloodstream, abdominal, and urinary tract infections. Infectious diseases in patients with dementia have already been ranked not only as the most common cause of hospital admission but also as a risk factor leading to a higher mortality rate [14,15,34,35]. Indeed, demented people usually have a great burden of risk factors for infections, such as reduced personal hygiene, an increased need for bladder catheterization, and dysphagia with an increased risk of inhalation and nasogastric tube feeding [36,37].

Among infections, pneumonia is certainly the most common cause of hospitalization in older adults, especially in those with dementia. This may be explained by dementia-related characteristics (e.g., dysphagia and impaired functional status) and more frequent use of sedative medications, which are all established risk factors for developing pneumonia [38]. Furthermore, several studies have reported that people with dementia tend to die more often from pneumonia [39,40]. In addition, a previous meta-analysis indicated that the odds of pneumonia-associated death were increased more than two-fold for patients with dementia than for those without dementia [15]. Conversely, in our cohort of demented patients, pneumonia was not significantly associated with an increased risk of in-hospital mortality. However, we found that the occurrence of respiratory failure in the demented patients increased three-fold their risk of death. These findings may be explained by a selection bias of patients with pneumonia, because, in our study, any acute medical condition resulting in respiratory failure was included in the so-called category “respiratory failure”, regardless of the etiology. Therefore, all the cases of pneumonia needing oxygen therapy, as well the other acute respiratory diseases—e.g., COPD exacerbation, acute heart failure, or pulmonary embolism—presenting with respiratory failure, contributed to the increased risk of death in the demented patients. Likewise, a retrospective observational cohort study including patients with dementia admitted to the Intensive Care Unit showed that the most frequent causes of death were respiratory [41].

It is well known that uncontrolled infections may lead to a bloodstream infection and, consequently, to acute organ dysfunction with an increased risk of death. As previously reported, older hospitalized patients are at high risk of developing bloodstream infections, and both advancing age and underlying comorbidities are associated with higher mortality [16,36,37,42]. However, although there is an extensive body of literature focused on the epidemiology of bloodstream infections, older patients, particularly those with dementia, have infrequently been the focus to date [37,43,44]. Furthermore, although advancing age and comorbidities are highly relevant for predicting the outcomes of patients diagnosed with bloodstream infections, in several previous studies, the differences in comorbidities were not taken into account in the statistical analysis [37]. In line with the previous reports [16,36,37,42], the present study showed that, compared with the non-demented patients, bloodstream infections were more common in those with dementia (7% vs. 10%). Furthermore, our analysis showed that, in the demented hospitalized patients, bloodstream infections increased by nearly 1.5 times the risk of in-hospital death. Similar results were also reported in a German population-based cross-sectional survey, in which Dasch et al. demonstrated that, in demented patients, the presence of infectious diseases, such as pneumonia and sepsis, was correlated with an increased likelihood of death in hospital [45].

Indeed, aging is associated with complex changes in the immune system, including an altered acute phase reaction with a greater risk of worse outcomes after severe infection [46], diagnostic difficulties, and potential differences in treatment and clinical quality. In addition, patients with dementia commonly have an atypical or asymptomatic clinical onset, and the febrile response may be blunted or absent, resulting in both increased risk of bloodstream infections and delayed diagnosis and treatment [47].

Therefore, our findings are of great interest for a better understanding of how to improve outcomes among older patients with dementia. These include appropriate ED management to reduce hospital admission, better access to clinical care, and tailored management of comorbidities and acute events to prevent poor clinical outcomes in this population.

## 5. Limitations

Our research has some limitations. Firstly, being a retrospective analysis, all the data were extracted from electronic records, and our classification of cases and diagnoses was based on the information available in the hospital records. In particular, the patients enrolled were classified as “demented” if they had received both an admission and discharge diagnosis of dementia in the hospital records. As a result, there may have been a misclassification of some demented patients, especially those with milder diseases. Moreover, a major limitation of this study was the lack of information about the sub-type of dementia. For this reason, it was not possible to assess the impact on mortality of the different subtypes of dementia. As a matter of fact, even though several studies addressing this issue have actually found no significant differences in mortality among dementia subtypes [48], a recent meta-analysis found that non-Alzheimer’s dementias were associated with higher mortality rates and shorter life expectancy than Alzheimer’s disease [49]. Furthermore, our ED has a dedicated geriatric unit for the early identification of frail patients, and our comprehensive assessment of older patients could be more accurate. As explained above about the reasons for the lack of data about the diagnosis of dementia and its subtypes, the ascertainment of acute medical conditions could have been incomplete.

Secondly, this is a single-center study, and the enrolled population could not be representative of all demented patients. Thirdly, as in prior studies, no information was available about the pharmacological treatments that are potential contributors to adverse clinical outcomes [50,51,52].

## 6. Conclusions

Among hospitalized older patients, dementia is associated with a higher risk of mortality. Furthermore, among older patients with dementia, the burden of comorbidities, any cause of respiratory failure, and bloodstream infections are independently associated with an increased risk of in-hospital mortality.

Therefore, among older patients with dementia, not only better access to care and adequate infection prevention to reduce hospital admission but also tailored management of both comorbidities and acute events, especially respiratory failure and bloodstream infections, should be encouraged to prevent poor outcomes.

## Figures and Tables

**Table 1 jcm-12-00059-t001:** Population demographics of all adults ≥ 65 years admitted to the Emergency Department (ED) and subsequently hospitalized.

Variable	All *n =* 48,962	Controls *n =* 45,402	Dementia *n =* 3559	*p*
Age ^§^	78 [72–84]	78 [71–84]	85 [80–89]	<0.001
Sex (Male)	25,352 (51.8)	23,948 (52.7)	1404 (39.4)	<0.001
Emergency Department presentation				
Dyspnea	9852 (20.1)	9096 (20.0)	729 (20.5)	0.525
Fever	9302 (19.0)	8520 (18.8)	782 (22.0)	<0.001
Chest pain	5978 (12.2)	5817 (12.8)	161 (4.5)	<0.001
Syncope	3775 (7.7)	3420 (7.5)	355 (10.0)	<0.001
Abdominal pain	5531 (11.3)	5324 (11.7)	207 (5.8)	<0.001
Diarrhea	1687 (3.4)	1553 (3.4)	134 (3.8)	0.279
Vomit	4008 (8.2)	3752 (8.3)	256 (7.2)	0.025
Malaise/fatigue	5785 (11.8)	5411 (11.9)	374 (10.5)	0.012
NEWS ≥ 5	7061 (14.4)	6533 (14.4)	528 (14.8)	0.470
Total ED time (hours)	16.8 [5.5–27.4]	16.2 [5.3–27.1]	21.3 [9.3–30.9]	<0.001
Clinical history				
Charlson Comorbidity Index	5 [4–7]	5 [4–7]	7 [5–8]	<0.001
Ischemic heart disease	10,950 (22.4)	10,305 (22.7)	645 (18.1)	<0.001
Congestive heart failure	9055 (18.5)	8159 (18.0)	895 (25.2)	<0.001
Peripheral vascular disease	10,764 (22.0)	9425 (20.8)	1338 (37.6)	<0.001
Previous stroke	5470 (11.2)	4635 (10.2)	834 (23.4)	<0.001
COPD ^#^	6539 (13.4)	5999 (13.2)	540 (15.2)	0.001
Diabetes	10,553 (21.6)	9781 (21.5)	772 (21.7)	0.842
Chronic liver disease	1739 (3.6)	1676 (3.7)	31 (0.9)	0.712
Chronic kidney disease	7963 (22.4)	7166 (15.8)	796 (22.4)	<0.001
Acute medical conditions				
Respiratory failure	6586 (13.5)	5904 (13.0)	681 (19.1)	<0.001
COPD exacerbation ^#^	1798 (3.7)	1651 (3.6)	147 (4.1)	0.132
Acute heart failure	5792 (11.8)	5172 (11.4)	619 (17.4)	<0.001
Acute myocardial infarction	2881 (5.9)	2798 (6.2)	83 (2.3)	<0.001
Acute atrial fibrillation	7310 (14.9)	6741 (14.8)	569 (16.0)	0.067
Pulmonary embolism	753 (1.5)	702 (1.5)	51 (1.4)	0.596
Ischemic stroke	5031 (10.3)	4549 (10.0)	482 (13.5)	<0.001
Hemorrhagic stroke	2172 (4.4)	1947 (4.3)	225 (6.3)	<0.001
Bloodstream infections	3456 (7.1)	3084 (6.8)	372 (10.5)	<0.001
Pneumonia	5119 (10.5)	4386 (9.7)	733 (20.6)	<0.001
Abdominal infection	1845 (3.8)	1743 (3.8)	102 (2.9)	0.003
Urinary tract infection	3756 (7.7)	3729 (7.2)	477 (13.4)	<0.001
Acute kidney injury	3337 (6.8)	2961 (6.5)	376 (10.6)	<0.001
Outcomes				
Death	5523 (11.3)	4865 (10.7)	666 (18.7)	<0.001
Discharge home	37,316 (76.2)	35,151 (77.4)	2165 (60.8)	<0.001
LOS ^£^	9.1 [5.5–14.8]	9.0 [5.5–14.6]	12.6 [6.1–15.5]	<0.001

^#^ Chronic Obstructive Pulmonary Disease; ^§^ Include percutaneous drainages, endoscopic operative procedures, and intravascular procedures; ^£^ Length of Hospital Stay.

**Table 2 jcm-12-00059-t002:** Population demographics of adults ≥ 65 years admitted to the Emergency Department (ED) and subsequently hospitalized. Comparison of patients with dementia and patients with no cognitive impairment (controls).

Variable	All *n =* 7118	Controls *n =* 3559	Dementia *n =* 3559	*p*
Age	84 [79–89]	84 [79–89]	84 [80–89]	0.866
Sex (Male)	2863 (40.2)	1459 (41.0)	1404 (39.4)	0.096
Emergency Department presentation				
Dyspnea	1438 (20.2)	709 (19.9)	729 (20.5)	0.287
Fever	1520 (21.4)	738 (20.7)	782 (22.0)	0.107
Chest pain	310 (4.4)	149 (4.2)	161 (4.5)	0.261
Syncope	716 (10.1)	361 (10.1)	355 (10.0)	0.422
Abdominal pain	397 (5.6)	190 (5.3)	207 (5.8)	0.204
Diarrhea	285 (4.0)	151 (4.2)	134 (3.8)	0.167
Vomit	499 (7.0)	243 (6.8)	256 (7.2)	0.289
Malaise/fatigue	746 (10.5)	372 (10.5)	374 (10.5)	0.485
NEWS ≥ 5 **	1062 (14.9)	534 (15.0)	528 (14.8)	0.434
Total ED time (hours) ***	23.1 [7.9–29.8]	21.6 [6.7–28.8]	21.3 [9.3–30.9]	<0.001
Clinical history				
Charlson Comorbidity Index	7 [5–9]	7 [5–9]	7 [5–8]	0.237
Ischemic heart disease	1530 (21.5)	885 (24.9)	645 (18.1)	<0.001
Congestive heart failure	1886 (26.5)	991 (27.8)	895 (25.2)	0.005
Peripheral vascular disease	2377 (33.4)	1039 (29.2)	1338 (37.6)	<0.001
Previous stroke	1445 (20.3)	611 (17.2)	834 (23.4)	<0.001
COPD ^#^	953 (13.4)	470 (13.2)	540 (15.2)	0.001
Diabetes	1537 (21.6)	765 (21.5)	772 (21.7)	<0.001
Chronic liver disease	106 (1.5)	75 (2.1)	31 (0.9)	<0.001
Chronic kidney disease	1724 (24.2)	928 (26.1)	796 (22.4)	<0.001
Acute medical conditions				
Respiratory failure	1377 (19.3)	696 (19.6)	681 (19.1)	0.337
COPD exacerbation ^#^	326 (4.6)	179 (5.0)	147 (4.1)	0.039
Acute heart failure	1270 (17.8)	651 (18.3)	619 (17.4)	0.169
Acute myocardial infarction	214 (3.0)	131 (3.7)	83 (2.3)	0.001
Acute atrial fibrillation	1246 (17.5)	677 (19.0)	569 (16.0)	<0.001
Pulmonary embolism	105 (1.5)	54 (1.5)	51 (1.4)	0.422
Ischemic stroke	960 (13.5)	478 (13.4)	482 (13.5)	0.459
Hemorrhagic stroke	400 (5.6)	175 (4.9)	225 (6.3)	0.006
Bloodstream infections	620 (8.7)	248 (7.0)	372 (10.5)	<0.001
Pneumonia	1215 (17.1)	482 (13.5)	733 (20.6)	<0.001
Abdominal infection	176 (2.5)	74 (2.1)	102 (2.9)	0.020
Urinary tract infection	740 (10.4)	263 (7.4)	477 (13.4)	<0.001
Acute kidney injury	731 (10.3)	355 (10.0)	376 (10.6)	0.217
Outcomes				
Death	1237 (17.4)	571 (16.0)	666 (18.7)	0.002
Discharge home	4628 (65.0)	2463 (69.2)	2165 (60.8)	<0.001
LOS ^£^	12.4 [6.1–15.3]	12.2 [6.0–15.1]	12.6 [6.1–15.5]	0.342

^#^ Chronic Obstructive Pulmonary Disease; ^£^ Length of Hospital Stay, ** National Early Warning Score; *** Emergency Department.

**Table 3 jcm-12-00059-t003:** Population demographics of adults ≥ 65 years admitted to the Emergency Department (ED) and subsequently hospitalized. Comparison with respect to all cause in-hospital death.

Variable	Survived *n =* 5881	Deceased *n =* 1237	*p* Value
Dementia	2893 (49.2)	666 (53.8)	0.002
Age	78 [72–84]	81 [75–87]	<0.001
Sex (Male)	2292 (39.0)	571 (46.2)	<0.001
Emergency Department presentation			
Dyspnea	1084 (18.4)	354 (28.6)	<0.001
Fever	1230 (20.9)	290 (23.4)	0.052
Chest pain	284 (4.8)	26 (2.1)	<0.001
Syncope	653 (11.1)	63 (5.1)	<0.001
Abdominal pain	339 (5.8)	58 (4.7)	0.074
Diarrhea	226 (3.8)	59 (4.8)	0.078
Vomit	422 (7.2)	77 (6.2)	0.129
Malaise/fatigue	626 (10.6)	120 (9.7)	0.175
NEWS ≥ 5 at ED admission **	701 (11.9)	361 (29.2)	<0.001
Total ED time (hours) ***	16.7 [5.6–27.3]	17.3 [5.3–28.4]	0.088
Clinical history			
Charlson Comorbidity Index	5 [4–7]	6 [5–8]	<0.001
Ischemic heart disease	1260 (21.4)	270 (21.8)	0.39
Congestive heart failure	1441 (24.5)	445 (36)	<0.001
Peripheral vascular disease	1936 (32.9)	441 (35.7)	0.035
Previous stroke	1176 (20)	269 (21.7)	0.089
Diabetes	1388 (23.6)	280 (22.6)	0.245
Chronic liver disease	86 (1.5)	20 (1.6)	0.381
Chronic kidney disease	1318 (22.4)	406 (32.8)	<0.001
Acute medical conditions			
Respiratory failure	784 (13.3)	593 (47.9)	<0.001
COPD exacerbation ^#^	293 (5)	33 (2.7)	<0.001
Acute heart failure	969 (16.5)	301 (24.3)	<0.001
Acute myocardial infarction	178 (3)	36 (2.9)	0.927
Acute atrial fibrillation	1051 (17.9)	195 (15.8)	0.077
Pulmonary embolism	80 (1.4)	25 (2)	0.091
Ischemic stroke	784 (13.3)	176 (14.2)	0.213
Hemorrhagic stroke	290 (4.9)	110 (8.9)	<0.001
Bloodstream infections	374 (6.4)	246 (19.9)	<0.001
Pneumonia	867 (14.7)	348 (28.1)	<0.001
Abdominal infection	162 (2.8)	14 (1.1)	<0.001
Urinary tract infection	663 (11.3)	77 (6.2)	<0.001
Acute kidney injury	507 (8.6)	224 (18.1)	<0.001
LOS ^£^	9.1 [5.6–14.5]	8.3 [3.3–17.9]	<0.001

^#^ Chronic Obstructive Pulmonary Disease; ^£^ Length of Hospital Stay; ** National Early Warning Score; *** Emergency Department.

**Table 4 jcm-12-00059-t004:** Multivariate analysis for the whole sample.

Variable	Hazard Ratio [95% Confidence Interval]	*p* Value
Dementia	1.13 [1.01–1.27]	0.033
Emergency Department presentation		
Charlson Comorbidity Index	1.03 [1.01–1.06]	0.048
NEWS ≥ 5 at ED admission **	2.43 [1.98–2.97]	<0.001
Acute medical conditions		
Respiratory failure	3.44 [3.05–3.90]	<0.001
COPD exacerbation ^#^	0.59 [0.41–1.05]	0.12
Acute heart failure	1.14 [0.99–1.3]	0.06
Acute myocardial infarction	1.13 [0.81–1.58]	0.46
Acute atrial fibrillation	0.9 [0.77–1.06]	0.22
Pulmonary embolism	0.7 [0.46–1.03]	0.07
Ischemic stroke	1.19 [0.99–1.42]	0.52
Hemorrhagic stroke	1.74 [1.41–2.16]	<0.001
Bloodstream infections	1.3 [1.12–1.5]	0.001
Pneumonia	1.01 [0.88–1.15]	0.92
Abdominal infections	0.6 [0.35–1.02]	0.06
Urinary tract infections	0.49 [0.39–0.62]	<0.001
Acute kidney injury	1.74 [1.5–2.02]	<0.001

^#^ Chronic Obstructive Pulmonary Disease; ** National Early Warning Score.

**Table 5 jcm-12-00059-t005:** Hazard ratio for in-hospital mortality in patients with dementia.

Variable	Hazard Ratio [95% Confidence Interval]	*p* Value
Emergency Department presentation		
Charlson Comorbidity Index	1.05 [1.01–1.1]	0.005
NEWS ≥ 5 at ED admission **	2.45 [1.88–3.2]	<0.001
Acute medical conditions		
Respiratory failure	3.08 [2.6–3.65]	<0.001
COPD exacerbation ^#^	0.72 [0.57–1.1]	0.1
Acute heart failure	1.05 [0.86–1.26]	0.68
Acute myocardial infarction	1.34 [0.85–2.13]	0.21
Acute atrial fibrillation	0.93 [0.75–1.16]	0.54
Pulmonary embolism	0.61 [0.33–1.11]	0.1
Ischemic stroke	1.01 [0.80–1.31]	0.91
Hemorrhagic stroke	1.84 [1.38–2.44]	<0.001
Bloodstream infections	1.41 [1.17–1.71]	0.001
Pneumonia	1.04 [0.88–1.24]	0.64
Abdominal infections	0.49 [0.22–1.1]	0.86
Urinary tract infections	0.5 [0.37–0.66]	<0.001
Acute kidney injury	1.64 [1.33–2.02]	<0.001

^#^ Chronic Obstructive Pulmonary Disease; ** National Early Warning Score.

## Data Availability

The data presented in this study are available on request from the corresponding author.

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
