# Peer review of "Outcomes and Predictors of In-Hospital Mortality among Older Patients with Dementia"

_jcm, 2022, doi:10.3390/jcm12010059_

Round 1

Reviewer 1 Report

The manuscript was well organised. The English language used was appropriate and professional. In summary, this is a retrospective study with a huge number of patients records. The data was well presented and the statistics used appropriate. It added to the literature by indicating that dementia is a determinant for the overall mortality in these patients. It would also complicate the morbidity of the patient’s health.

I have no further comments and leave it to the editors to decide if they want to publish the work.

Author Response

REVIEWER 1 COMMENT (Round 1)

The manuscript was well organised. The English language used was appropriate and professional. In summary, this is a retrospective study with a huge number of patients records. The data was well presented and the statistics used appropriate. It added to the literature by indicating that dementia is a determinant for the overall mortality in these patients. It would also complicate the morbidity of the patient’s health.

I have no further comments and leave it to the editors to decide if they want to publish the work.

RESPONSE:

We thank the Reviewer for taking the necessary time and effort to review the manuscript. We sincerely appreciate the valuable comments.

Reviewer 2 Report

The manuscript entitled 'Outcomes and predictors of in-hospital mortality among older patients with dementia' by De Matteis et al. is a well structured manuscript with a clear rationale and methodology. Results are presented in detail and adequately discussed. Overall, the submitted manuscript is up to date with interesting findings that are well described.

Author Response

REVIEWER 2 COMMENT (Round 1)

The manuscript entitled 'Outcomes and predictors of in-hospital mortality among older patients with dementia' by De Matteis et al. is a well structured manuscript with a clear rationale and methodology. Results are presented in detail and adequately discussed. Overall, the submitted manuscript is up to date with interesting findings that are well described.

RESPONSE:

We thank the Reviewer for taking the necessary time and effort to review the manuscript. We sincerely appreciate the valuable comments.

Reviewer 3 Report

Taking into account the increasing number of individuals with dementia, studies investigating dementia and premature mortality are needed. The manuscript is well-written, interesting, and well-research. Below are my comments:

1. The term “elderly” is ageist and many gerontological journal asked authors not to use the words. I encourage the Authors to read Dale et al. (2011). Use of the Term “Elderly”. Journal of Geriatric Physical Therapy, 34 (4), 153-154. doi: 10.1519/JPT.0b013e31823ab7ec

2. There are many different types of dementia (e.g., frontotemporal dementia, dementia with Lewy bodies, Alzheimer’s  disease), which diagnosis did the patients receive? The type is a very important variable that impacts mortality. Please see other studies that confirm it:

Garcia-Ptacek, S., Farahmand, B., Kåreholt, I., Religa, D., Cuadrado, M. L., & Eriksdotter, M. (2014). Mortality risk after dementia diagnosis by dementia type and underlying factors: a cohort of 15,209 patients based on the Swedish Dementia Registry. Journal of Alzheimer's disease : JAD, 41(2), 467–477. https://doi.org/10.3233/JAD-131856

Liang, C. S., Li, D. J., Yang, F. C., Tseng, P. T., Carvalho, A. F., Stubbs, B., Thompson, T., Mueller, C., Shin, J. I., Radua, J., Stewart, R., Rajji, T. K., Tu, Y. K., Chen, T. Y., Yeh, T. C., Tsai, C. K., Yu, C. L., Pan, C. C., & Chu, C. S. (2021). Mortality rates in Alzheimer's disease and non-Alzheimer's dementias: a systematic review and meta-analysis. The Lancet. Healthy longevity, 2(8), e479–e488. https://doi.org/10.1016/S2666-7568(21)00140-9

I understand that the Authors included lack of knowledge about the sub-type, this this should be also stated earlier in the text as this is a major limitation.

Author Response

Response to the Reviewers

Taking into account the increasing number of individuals with dementia, studies investigating dementia and premature mortality are needed. The manuscript is well-written, interesting, and well-research. Below are my comments:

COMMENT 1

The term “elderly” is ageist and many gerontological journal asked authors not to use the words. I encourage the Authors to read Dale et al. (2011). Use of the Term “Elderly”. Journal of Geriatric Physical Therapy, 34 (4), 153-154. doi: 10.1519/JPT.0b013e31823ab7ec

RESPONSE 1

We thank the Reviewer for this comment. As suggested in the revised version of the manuscript we replaced the ageist term “elderly” with “older” (pag. 1, line 43, and pag.2, line 63).

COMMENT 2

There are many different types of dementia (e.g., frontotemporal dementia, dementia with Lewy bodies, Alzheimer’s  disease), which diagnosis did the patients receive? The type is a very important variable that impacts mortality. Please see other studies that confirm it:

Garcia-Ptacek, S., Farahmand, B., Kåreholt, I., Religa, D., Cuadrado, M. L., & Eriksdotter, M. (2014). Mortality risk after dementia diagnosis by dementia type and underlying factors: a cohort of 15,209 patients based on the Swedish Dementia Registry. Journal of Alzheimer's disease : JAD, 41(2), 467–477. https://doi.org/10.3233/JAD-131856

Liang, C. S., Li, D. J., Yang, F. C., Tseng, P. T., Carvalho, A. F., Stubbs, B., Thompson, T., Mueller, C., Shin, J. I., Radua, J., Stewart, R., Rajji, T. K., Tu, Y. K., Chen, T. Y., Yeh, T. C., Tsai, C. K., Yu, C. L., Pan, C. C., & Chu, C. S. (2021). Mortality rates in Alzheimer's disease and non-Alzheimer's dementias: a systematic review and meta-analysis. The Lancet. Healthy longevity, 2(8), e479–e488. https://doi.org/10.1016/S2666-7568(21)00140-9

I understand that the Authors included lack of knowledge about the sub-type, this this should be also stated earlier in the text as this is a major limitation.

RESPONSE 2

As suggested, in the revised version of the manuscript we stated earlier in the text, in the methods section, the lack of knowledge about the sub-type of dementia (pag.2, Lines 79-81).

Similarly, we expanded on this concern in the limitations section (pages 11-12, Lines 354-360).

Moreover, as suggested, we included in the references the suggested articles about this issue (references 48 and 49).